# An Experimental Investigation on Polarization Process of a PZT-52 Tube Actuator with Interdigitated Electrodes

**DOI:** 10.3390/mi13101760

**Published:** 2022-10-18

**Authors:** Yonggang Liu, Aoke Zeng, Shuliang Zhang, Ruixiang Ma, Zhe Du

**Affiliations:** 1School of Mechatronics Engineering, Henan University of Science and Technology, Luoyang 471003, China; 2Collaborative Innovation Center of Machinery Equipment Advanced Manufacturing of Henan Province, Henan University of Science and Technology, Luoyang 471003, China; 3Luoyang Mining Machinery Engineering Design Institute Co., Ltd., Luoyang 471003, China

**Keywords:** piezoceramic tube actuator, interdigitated electrodes, electric field analysis, polarization process, axial displacement

## Abstract

The manipulator is the key component of the micromanipulator. Using the axial expansion and contraction properties, the piezoelectric tube can drive the manipulator to achieve micro-motion positioning. It is widely used in scanning probe microscopy, fiber stretching and beam scanning. The piezoceramic tube actuator used to have continuous electrodes inside and outside. It is polarized along the radial direction. There are relatively high polarization voltages, but poor axial mechanical properties. A new tubular actuator is presented in this paper by combining interdigitated electrodes and piezoceramic tubes. The preparation, polarization and mesoscopic mechanical properties were investigated. Using Lead Zirconate Titanate (PZT-52) as a substrate, the preparation process of interdigitated electrodes by screen printing was studied. For initial polarization voltage determination, the local characteristic model of the actuator was extracted and the electric field was analyzed by a finite element method. By measuring the actuator’s axial displacement, we measured the actuator’s polarization effect. Various voltages, times and temperatures were evaluated to determine how polarization affects the actuator’s displacement. Optimal polarization conditions are 800 V, 60 min and 150 °C, with a maximum displacement of 0.88 μm generated by a PZT-52 tube actuator with interdigitated electrodes. PZT-52 tube actuators with a continuous electrode cannot be polarized under these conditions. The maximum displacement is 0.47 μm after polarization at 4 kV. Based on the results, the new actuator has a more convenient polarization process and a greater axial displacement from an application standpoint. It provides technical guidance for the preparation and polarization of the piezoceramic tube actuator. There is potential for piezoelectric tubular actuators to be used in a broader range of applications.

## 1. Introduction

Piezoceramics are functional information materials that can convert mechanical energy and electrical energy into each other [1,2]. There are three properties: piezoelectricity, dielectricity and the elastic property. Furthermore, they are widely used in medical imaging [3,4], sensors [5], actuators [6], ultrasonic motors [1,7,8], micro-electro-mechanical systems [9] and ultra-precision measurements [10,11]. With piezoelectric actuators, small displacements are produced by the inverse piezoelectric effect of piezoelectric materials. They also have the advantages of a simple structure, fast response, high controllable precision, low energy consumption, small size and flexible combination [12,13]. In response to the demands of different fields, piezoceramic sensors and actuators have evolved rapidly. Various piezoceramic actuators with different shape specifications have been produced one after another, such as thin sheets, cylindrical, tubes, strips and so on [14,15,16,17]. High-precision mechanical engineering, biomedical science, aerospace and other micro-drive fields need high-quality actuators with both good electromechanical characteristics and low preparation costs [18,19,20,21].

Domain engineering methods have been adopted to improve the properties of piezoelectric materials in nanoscale. The piezoelectric, ferroelectric and dielectric properties of piezoelectric materials have been studied in the past based on the domain size, walls, orientation and interdomain spacing. Kathavate et al. found that the differences in nano-mechanical and piezoelectric properties can be attributed to the changes in domain configurations. The annealing treatment caused significant changes in the domain configurations, thereby leading to enhancements in mechanical properties. However, the piezoelectric constant *d*_33_ decreases with an increase in annealing temperature [22,23]. They researched the effect of dopants on the “hard” and “soft” behavior originating from the polarization and depolarization behavior and the mobility of domains. The dopants were adopted to improve the electromechanical properties of PZT [24]. Ren [25] provided a reversible domain switching method to achieve a large electro-strain effect. It could cause an aged BaTiO_3_ single crystal to generate a large recoverable nonlinear strain of 0.75% at a low field of 200 V mm^−1^. Höfling et al. [26] developed a method for mechanically imprinting dislocation networks. It yielded a mechanical restoring force to revert electric-field-induced domain wall displacement on the macroscopic level and high pinning force on the local level. This induced a huge increase in the dielectric and electromechanical response at intermediate electric fields in barium titanate (electric-field-dependent permittivity, *ε*_33_, is 5800 and large-signal piezoelectric coefficient, *d*_33_, is 1890 picometers V^−1^).

The mechanical properties of bulk ceramics are related to both the piezoelectric materials and structure. New functional and different forms of piezoceramic actuators are being developed rapidly. The piezoceramic tube actuator can drive the micromanipulator to achieve mesoscale micro-positioning. Zhang et al. [27] researched the ratio of the outer radius to the inner radius effected on the effective piezoelectric constant of the piezoelectric tube composite actuator. A piezoelectric transducer with the effective piezoelectric tensile constants having the same sign was made. Based on the results of analyses carried out with ANSYS finite element software, Fan et al. [28]. proposed an intelligent control scheme to evaluate the maximum displacement, output and response frequencies of a piezoceramic tube actuator. They also analyzed an intelligent damper using the tube actuator. The dynamic mechanical properties of the intelligent damper were analyzed, and the constitutive relationship under the sinusoidal simple harmonic action and the calculation formula of the energy consumed in one cycle was obtained. Raghuvanshi et al. [29]. researched methods for driving piezoelectric tube scanners through the design of an electrode driving mode. The effects of the electrode on the lateral and vertical scan range, vertical displacement, tilting and vertical cross-coupling were obtained in the experiment. Wang et al. [30] adopted the plane strain theory to derive the electromechanical equivalent circuits of the radial vibration of a piezoceramic and metal long tube. The radial resonance/anti-resonance frequency equation and the expression of the effective electromechanical coupling coefficient were obtained. The effects of the radial geometry dimension of the transducer on the vibration characteristics were analyzed.

Compared with other types of piezoceramic actuators, interdigitated electrode piezoceramic actuators have obvious anisotropy, a small size and large output displacement. Additionally, they can generate strong driving forces in a specific direction [31,32,33]. We have previously investigated a type of piezoceramic tube actuator with interdigitated electrodes using ABAQUS finite element software. The influence of the electrode shape and structural dimensions on the clamping stress and free strain of some models was analyzed. The results showed that the actuator can achieve 2.75 times the axial stress and 1.55 times the axial free strain of the actuator with a continuous electrode (inner and outer cylindrical surface electrode) [34]. Nanoscale domain engineering provides an effective means to improve the properties of thin films and piezoelectric materials. The mechanical properties of bulk ceramics are related to both the electromechanical properties and the polarization of piezoelectric materials. It is necessary to reveal the deformability of piezoelectric actuators at micrometer and submicrometer scales by means of macroscale deformation experiments. The preparation of the actuator whose outer circular surface is covered with full interdigital electrodes is relatively difficult. A piezoceramic tube actuator with 180-degree interdigitated electrodes is proposed in this paper. We provide a detailed description of the electrode printing, and an experiment has been conducted to polarize the samples. The effects of the polarization process on actuator displacements is analyzed.

## 2. Sample Preparation and Polarization Experiment

### 2.1. Structure and Preparation of Tube Actuator with PZT-52

A schematic of the piezoceramic tube actuator with interdigitated electrodes is shown in Figure 1. The surface of the actuator is covered with equally spaced opposite electrodes arranged similarly to crossed fingers [34]. This tube is 25 mm in length, 17.5 mm in inner diameter and 20.5 mm in outer diameter. The electrode covers a 180-degree cylindrical surface, its branches are 1 mm wide, and its branches are separated by 0.6 mm. In total, there are thirteen branch electrodes. This structure makes full use of the surface area of the actuator. The direction of polarization is along the length direction. In addition, it can take advantage of the high piezoelectric constant, *d*_33_, which would translate into a large driving displacement axially.

Lead Zirconate Titanate (PZT-52) is commonly used as an actuator material due to its high dielectric constant and piezoelectric constant. The raw materials were Pb(H_2_COO)_2_-3H_2_O (the purity is 99.5%), Ti((CH_3_)_2_CHO)_4_(98%), ZrO_2_ and C_6_H_8_O_7_H_2_O (99.5%). Pb(Zr_o_._52_Ti_o_._48_)O_3_ was used at the stoichiometric ratio, and dopants such as La3+, Sb5+, Bi were added as well. This is a relatively mature technology used to prepare P-52 tubes via the solid phase method, and the preparation process generally involves ingredients—mixed grinding—pre-firing—second grinding—granulation—molding—plastic—sintering into porcelain—shape processing. Figure 2 shows the samples prepared at YU Hai Electronic Ceramics Co., Ltd. (Zibo, China).

The piezoceramic tube actuator with continuous electrodes is based on a ceramic tube with electrode layers attached on the inner and outer surfaces, as shown in Figure 3. The dielectric constant, $\varepsilon_{33}^{T}$, is 2400. The piezoelectric constant, *d*_31_, is 204 × 10^−12^ C N^−1^, and *d*_33_ is 520 × 10^−12^ C N^−1^. The volume density is 7.6 × 10^3^ kg m^−3^. The Curie point is 270 °C. The Young’s modulus is 59 × 10^9^ N m^−2^. The Poisson ratio is 0.36. Generally, the inner is a positive electrode and the outer is a negative electrode. The polarization direction is radial. Due to the large aspect ratio of the tube actuator, when the voltage difference on the electrode surface changes, the actuator mainly displays expansion and contraction along the axial direction.

### 2.2. Electrode Printing with Silver

Piezoceramics are non-metallic materials. The samples are not electrically conductive. In order to supply power to the actuator, the layer of metal electrodes must be plated on the surface of the sample at the positive and negative electrode positions in the working electrode area. The electrode material should have good electrical conductivity and be tightly and firmly bonded to the surface of the actuator. The electrode material is conductive silver glue, which contains 60–70% silver (weight ratio) and has a particle size of 3–5 μm, cured at 150 °C.

The silver electrode material is one of the most important common materials for producing piezoceramic actuators, mainly due to its high conductivity, low cost and stability. There are many types of electrode preparation methods for piezoelectric components, such as the electrode deposition method, photolithography processing method and laser processing method. The screen-printing method was adopted in this study. Compared with the other methods, screen-printing for some thin, round or piezoceramic tube actuators can be easily carried out in a laboratory. Moreover, the research on small batch production indicates that it is not only a simple production process, but is also low-cost and has high efficiency.

Electrode patterns should be printed on sulfuric acid paper in the size of the interdigital electrodes. The electrode printing screen is shown in Figure 4. The electrode area was accurately placed on the surfaces of the piezoceramic tube samples, and the silver paste was evenly applied to the surface of the actuator with a squeegee. Then, the samples were placed in a 10 A chamber resistance furnace and heated for 30 min to allow the electrode to closely adhere to the surface of the sample, as shown in Figure 5.

In the process of producing the silver electrodes, it was important to ensure that the screen and the samples were completely attached. The electrode layer should be coincident with the ceramic tube, to prevent the electrode from being distorted or leaking silver after printing, which would affect the performance of the actuator. The samples of the piezoceramic tube actuator are shown in Figure 6.

### 2.3. Thermal Polarization Experiment

#### 2.3.1. Electric Field Analysis

A piezoceramic actuator cannot generate its piezoelectric effect until it is polarized [1]. Before polarization, the conditions should be determined in advance. The polarization process mainly includes the voltage, temperature and time. Temperature and time need to be obtained through experimental research. The polarization voltage is related to the structure of the actuator and can be preliminarily determined by electric field analysis.

Due to the periodic arrangement of branch electrodes along the axial direction, the actuator is polarized in the axial direction. The actuator’s polarization axis is perpendicular to the branch electrodes’ central plane. It shows the opposite polarization direction on each side. The electric field is circularly symmetric along the axial direction of the sample. To simplify the model, the partial model between two adjacent opposite electrodes was adopted. The height of the model was 3.2 mm, as shown in Figure 7.

The partial models were built using finite element analysis software, ANSYS (Version 2019 R2, ANSYS, Inc., Canonsburg, PA, USA). SOLID98 ELEMENT was adopted for electric field analysis, with 84,103 nodes and 62,345 units. The “0 V” electrode area was set to zero potential, and the “+” electrode area to high volts. A vector diagram of the axial electric field is shown in Figure 8.

As shown in Figure 8, the electric field lines parallel to the axial direction of the actuator are arc-shaped, and they are mirror images of the center plane of the branch electrodes. There is non-uniformity and non-linearity in the electric field in the electrode attachment area. It is common for the electric fields to be uniform between branch electrodes. They are sparse in the area beneath the electrode layer.

There is very low electric field strength in the electrodeless area, where the maximum electric field strength appears at the area marked “MAX”. Additionally, the electrode edge area where the “MAX” is located is easily broken down in the polarization experiment. Near the electrodes, the electric field lines are relatively concentrated, and the electric field intensity is relatively high. However, in the middle area of the two branch electrodes, the electric field intensity is relatively low. It is necessary to select polarization voltages based on the voltage at which the maximum axial field strength is observed, in order to avoid breakdown when the sample is polarized under a high DC voltage.

As a comparison, the electric field of the piezoceramic tube actuator was also analyzed with a continuous electrode. The variation in the maximum electric field strength with the applied voltage is shown in Figure 9. Calculations of electromagnetic fields are based on Maxwell’s governing equations. An electrostatic field analysis via the finite element method can be simplified as the Poisson equation of electric potential:(1)$$d\varphi =-\vec{E}d\vec{l}$$
where $d\vec{l}$ is the distance between two points, d*φ* is the potential difference between two points, and $\vec{E}$ is the electric field vector. Thus, the maximum electric field strength of the sample is linearly related to the loading voltage. A piezoelectric tube actuator with a continuous electrode is covered with electrodes on both the inside and outside. It acts as a capacitor along the wall thickness and has uniform electric field strength, which is the voltage to wall thickness ratio. The electric field strength also increases linearly with the increase in voltage.

The polarization electric field of PZT piezoceramics is generally 3~5 kV/mm, and the breakdown electric field strength is 5~7 kV/mm. The principle of polarization electric field setting for PZT piezoelectric ceramics is as follows [1,35]:(2)$$E_{p}=(3\sim 4)E_{o}$$
(3)$$E_{p}<E_{b}$$
where *E_p_* is the polarization electric field; *E*_o_ is the coercivity electric field; *E*_*b*_ is the dielectric breakdown electric field.

Upon applying 800 V to the sample, the maximum electric field strength reaches the upper limit of the dielectric breakdown strength of the PZT piezoelectric ceramics used, 6.98 kV/mm, as shown in Figure 9. Nevertheless, the electric field strength of the piezoceramic tube actuator with a continuous electrode only reaches 0.8 kV/mm under the same loading conditions. Therefore, the piezoceramic tube actuator with interdigitated electrodes is characterized by its ability to fully polarize at lower polarization voltages. This can effectively improve the polarization efficiency and further reduce the risk of high-voltage power application.

#### 2.3.2. Polarization Experiment

For samples to exhibit piezoelectric behavior, all the domains need to be aligned in one direction by polarization [36]. The thermoelectric polarization method was chosen in this study, which can polarize the actuators below the Curie temperature. It has the characteristics of a wide polarization temperature range, high insulation strength and good moisture resistance. The actuator polarization equipment is shown in Figure 10.

In order to investigate the output axial displacement of the actuators under different polarization processes, the appropriate polarization voltage could be determined using the results of the electric field analysis. However, the polarization time and polarization temperature should be initially selected based on experience. The selected polarization process conditions are shown in Table 1.

After the polarization conditions were determined, the wires were welded in the positive and negative electrode areas of the sample by welding. In this way, the voltage is easy to load in polarization. The sample was checked with a multimeter to ensure that the welded samples were in good contact and that the silver electrode area was highly conductive.

Methyl silicone oil was added to the oil bath box and heated to the polarization temperature. The samples were placed in the oil bath box and completely immersed in the silicone oil for approximately 10 min to allow the sample to reach the polarization temperature. Then, the positive and negative electrodes of the sample were connected to the high-voltage power supply. The loading voltage was increased step by step to the polarization voltage. The samples were kept at the polarization voltage and polarization temperature for 30 min to ensure complete polarization.

The samples should be removed from the high-voltage power supply as soon as possible, dipped in methyl silicone oil and left at room temperature after polarization is complete. After rapidly loading the voltage below the polarization voltage, the samples were cooled to room temperature. Then, we turned off the high-voltage power supply. The polarized samples were removed, cleaned with kerosene and finally scrubbed with drug cotton. They were placed at room temperature to dry naturally and kept for 12 h before performance testing. The same steps were repeated for the test, and different polarization conditions were selected for the actuators. Polarization was not possible if there was a breakdown during polarization.

The output displacement of the actuator is one of the most important performance parameters. It is influenced by the residual polarization of the actuator. In this work, the residual polarization of the actuator was tested by detecting the axial displacement of the samples.

## 3. Displacement Test and Analysis

The displacement detection system in document [35] was used to investigate the output displacements of the actuator. The bottom of the actuator and laser displacement sensor were fixed on the glass and placed on the damping table for noise reduction. The laser head of the sensor was positioned facing the center of the electrode on the end face of the sample, as shown in Figure 11.

### 3.1. The Influence of Polarization Voltage on Axial Displacement

The excitation applied to the piezoceramics in the experiment was a sinusoidal excitation signal of 0~200 V at 1 Hz, amplified by high-voltage amplifiers. Different polarization voltages (500 V, 600 V, 700 V, 750 V, 800 V) were applied to the sample when the polarization time was 20 min and the polarization temperature was 110 °C. The output displacement time domain curves and the response curves of the samples were measured, as shown in Figure 12. There were some nonlinear fluctuations in the curves, primarily due to domain reorientation. The PZT-52 sample had defects such as porosity and impurities due to sintering [36]. The nonlinear contribution below the Curie temperature was primarily due to the motion of the domain walls and their interactions with defects [37]. The ferroelectric domain configurations and defect vacancies can be controlled by the addition of aliovalent dopants in PZT-52, which causes a respective increase in the polarization and *d*_33_ values [38].

Figure 12 shows that the axial displacement of the sample increased with increasing polarization voltage under the same experimental conditions. As the polarization voltage increased from 750 V to 800 V, there was a significant decrease in the actuator’s maximum driving displacement. This was due to the fact that when the polarization voltage reached approximately 800 V, the sample reached the maximum electric field strength. It is likely that the actuator would be damaged if the voltage exceeds 800 V during polarization, since the electric field strength would exceed the piezoceramic’s maximum breakdown voltage.

At the polarization voltage of 500 V, the internal electric field strength of the piezoceramics was too small. Consequently, the internal electric domain of the actuator could not be ordered along the direction of the electric field, resulting in a sample that was not properly polarized. It was estimated at this time that the maximum driving displacement of the sample was 0.12 μm, which was approximately half the maximum polarization voltage.

### 3.2. The Influence of Polarization Time on Axial Displacement

The polarization time is the holding time required for a piezoceramic to change from one equilibrium state to another. It is important to study the effect of different polarization times on the driving displacement of the piezoceramic sample in order to achieve a fully polarized sample. Our samples were polarized using a polarization voltage of 500 V, a polarization temperature of 110 °C and different polarization times (20 min, 30 min, 40 min, 50 min, 60 min). We detected the displacement of samples under the same loading signal source. The output displacement time-domain curve and the corresponding curve of the electric ceramic are shown in Figure 13.

As can be seen in Figure 13, the driving displacement of the sample increases with increasing polarization time. The alignment degree of the electric domain inside the actuator is too low to reflect the driving performance when the polarization time changes over a period of 20–40 min due to the short polarization time. The maximum axial driving displacement can be reflected in the figure, which remains the same. Generally, the longer the polarization time, the more fully polarized the sample is, and the better the driving displacement. It is important to note that after reaching a certain polarization time, the internal electric domain orientation arrangement of the sample was complete, and extending the polarization time has little effect on its displacement. This actuator achieves a good polarization effect when the polarization time reaches approximately 60 min, and the maximum displacement of the actuator can reach 0.30 μm at this time.

### 3.3. The Influence of Polarization Temperature on Axial Displacement

In the polarization process conditions, the polarization temperature is another important index affecting the performance of piezoceramics. To study the displacement of the actuator in relation to the polarization temperature, a polarization voltage of 500 V, a polarization time of 20 min and a polarization temperature of 110 °C, 120 °C, 130 °C, 140 °C and 150 °C were used to polarize the samples firstly. The output displacement time-domain curves and corresponding curves of the sample were obtained, as shown in Figure 14.

Figure 14 shows that the axial displacement gradually increases as the polarization temperature increases. Thermal motion within piezoceramics is more active at higher polarization temperatures, at a specific polarization voltage and polarization time. In general, the lower the crystal anisotropy with temperature, the easier it is to arrange and polarize the electric domains. When the polarization temperature was 110 °C, the actuator was not sufficiently polarized, which showed a small axial displacement, and it did not achieve good driving performance, as shown in the figure. Moreover, when the polarization temperature exceeded 150 °C, the maximum displacement of the sample reached a stable state. A higher polarization temperature can generally shorten the polarization time and improve the polarization efficiency. However, if the temperature is too high, the actuators would be broken down due to various factors such as actuator manufacturing errors.

Combined with Figure 12, Figure 13 and Figure 14, it can be seen that the axial displacement of the actuator shows a significant growth trend with the increase in polarization voltage, polarization time and polarization temperature under the same loading conditions. This indicates that this new actuator preparation and polarization method can obtain good driving performance. As can be seen from the graph, displacement fluctuates under different polarization conditions. The following experimental errors are mainly responsible for this. Firstly, the samples may display poor densification and adulteration with other impurities, which affects the later polarization process. Secondly, due to a lack of precision in the preparation process, the silver electrode structure of the sample deviates from the ideal situation during manual printing. Lastly, it is necessary to consider the environmental noise during displacement measurement, which is introduced into the experiment by precision sensors.

The axial displacement of the sample is shown in Figure 15: here, the polarization voltage was 800 V, the polarization time was 60 min, and the polarization temperature was 150 °C. A PZT-52 tube actuator with a continuous electrode was also polarized under the same conditions. However, its axial displacement was not detectable. In order to polarize the tube actuator with a continuous electrode, it was heated to 150 °C for 40 min under 4 kV voltage. In Figure 15, the result of the axial displacement experiment is presented for the actuator with interdigitated electrodes. Obtaining good polarization at relatively low voltages is of the utmost importance from a functionality point of view. As can be seen from the figure, with the increase in driving voltage, the axial displacement gradually increases. Actuators with interdigitated electrodes can reach 0.88 μm, while actuators with continuous electrodes can reach 0.47 μm. When the driving voltage decreases, the output displacement does not return by the original path, but the phenomenon of displacement hysteresis appears. The overall output displacement shows a circular shape, which is due to the hysteresis characteristics of the piezoceramic itself.

## 4. Conclusions

In this paper, a new type of piezoceramic tube actuator with a 180-degree interdigitated electrode layer was proposed. PZT-52 was used to prepare samples of 25 mm length, 17.5 mm inner diameter and 20.5 mm outer diameter. Screen printing was used to fabricate the actuators, using interdigitated electrode patterns printed on sulfuric acid paper with 1 mm width and 0.6 mm spacing. We polarized the samples using thermal polarization. The polarization voltage had to be determined preliminarily. For analysis of the electric field by the finite element software ANSYS, a partial characteristic model of the actuator was developed based on the structural characteristics of the actuator. Moreover, a piezoceramic tube actuator with the same profile but with a continuous electrode was analyzed for comparison. Based on the results of the electric field analysis, the polarization voltage of the sample was preliminarily determined. A variety of polarization voltages, times and temperatures were used to polarize the samples. Displacement experiments were conducted to determine the influence of the polarization voltage, polarization time and polarization temperature on axial displacement. The samples were fully polarized under the conditions of 800 V voltage, 60 min and 150 °C, but the piezoceramic actuators with continuous electrodes of the same size could not be polarized. This shows that the actuator’s polarization process is more effective. It is possible to achieve a maximum displacement of 0.88 μm when loaded with a 1 Hz and 0~200 V sinusoidal excitation signal.

## Figures and Tables

**Figure 1 micromachines-13-01760-f001:**
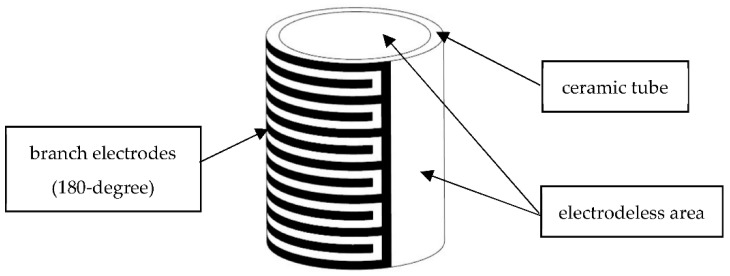
The schematic of piezoceramic tube actuator.

**Figure 2 micromachines-13-01760-f002:**
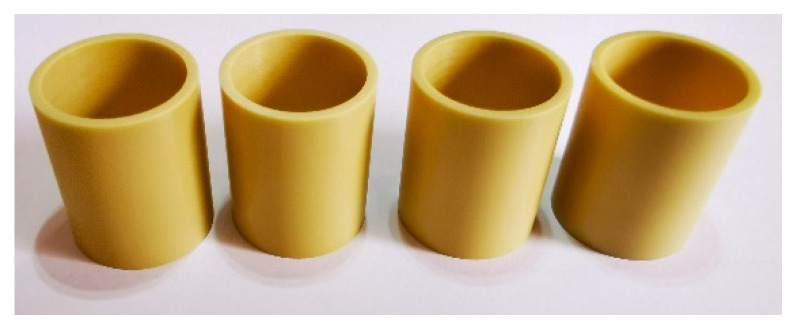
The samples of PZT-52 tube actuator.

**Figure 3 micromachines-13-01760-f003:**
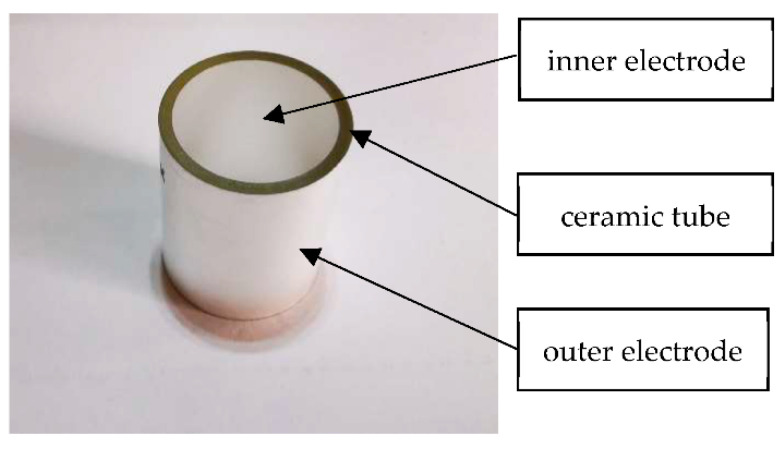
The PZT-52 tube actuator with continuous electrode.

**Figure 4 micromachines-13-01760-f004:**
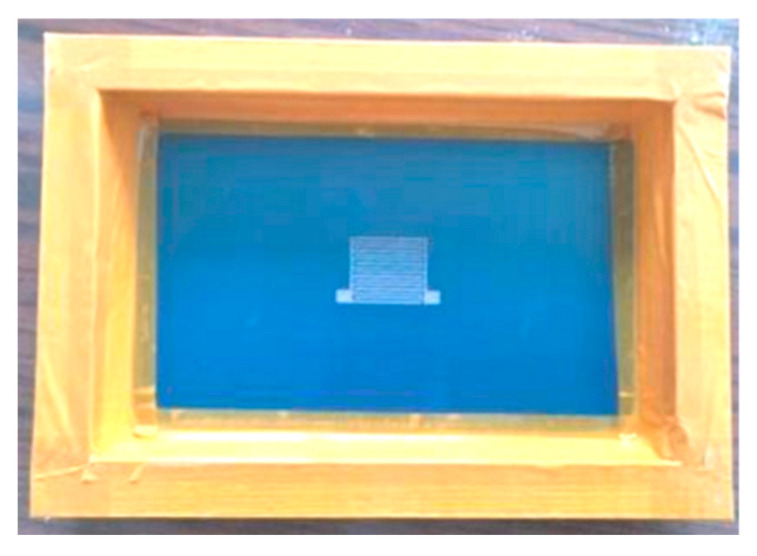
Interdigitated electrode printing screen.

**Figure 5 micromachines-13-01760-f005:**
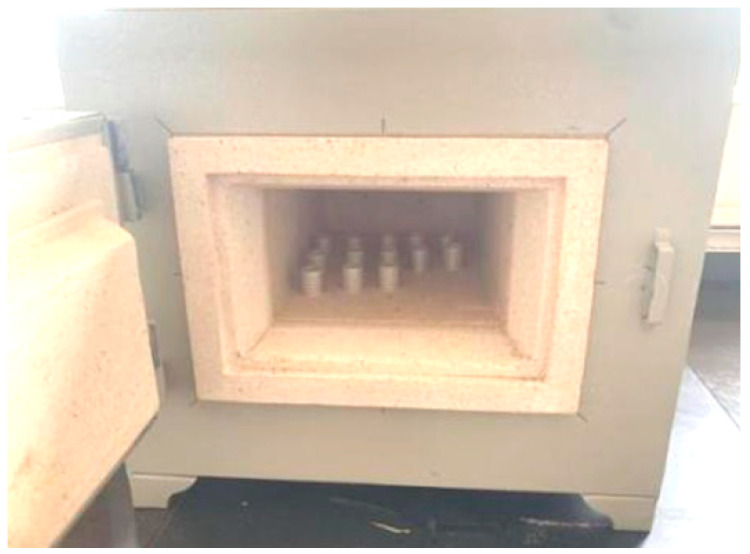
Heat-cured silver electrodes.

**Figure 6 micromachines-13-01760-f006:**
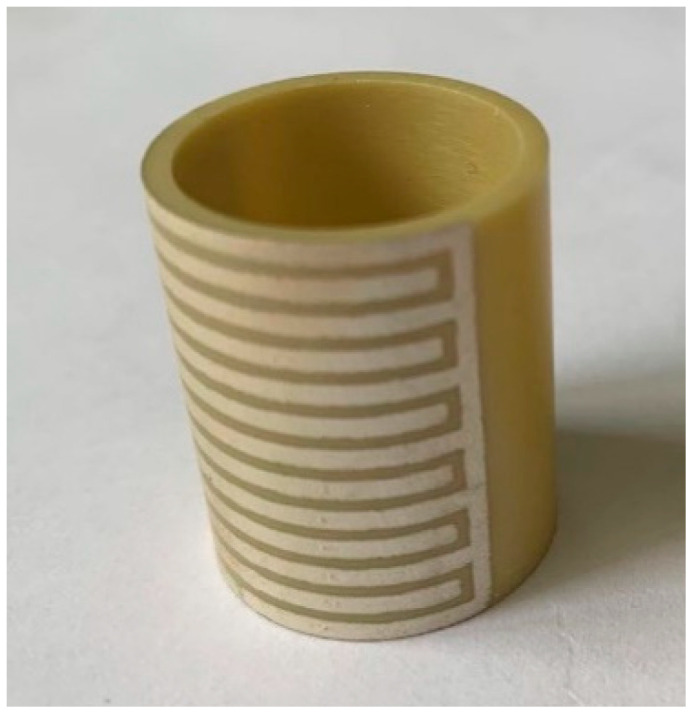
The samples of the piezoceramic tube actuator with interdigitated electrodes (180-degree).

**Figure 7 micromachines-13-01760-f007:**
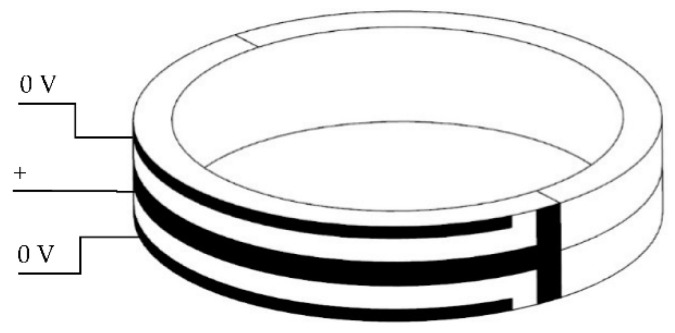
Partial model with two pairs of electrodes.

**Figure 8 micromachines-13-01760-f008:**
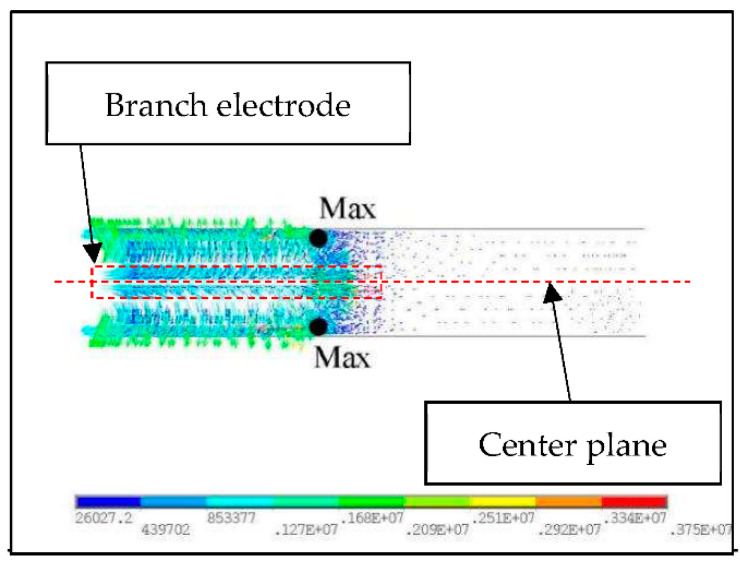
The electric field of the partial model.

**Figure 9 micromachines-13-01760-f009:**
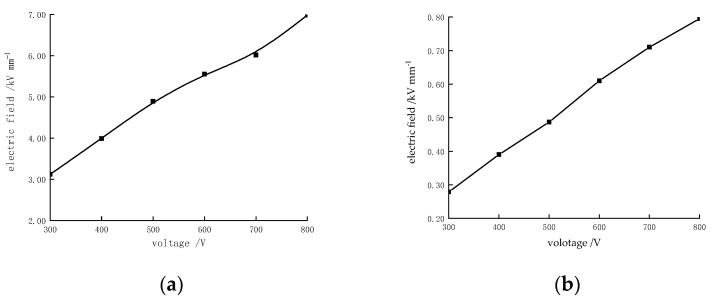
Electric field strength varies with voltage. (**a**) Partial model; (**b**) piezoceramic tube actuator with continuous electrode.

**Figure 10 micromachines-13-01760-f010:**
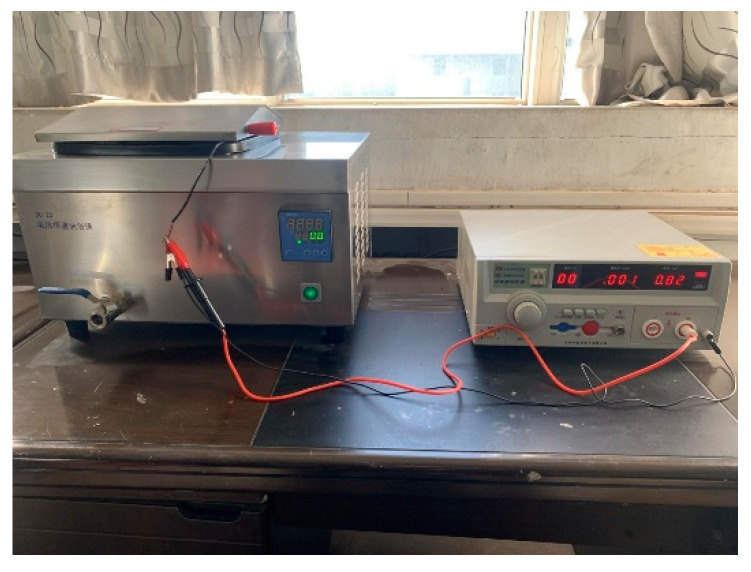
Polarization equipment: temperature-controlled oil bath box, high-voltage DC power supply.

**Figure 11 micromachines-13-01760-f011:**
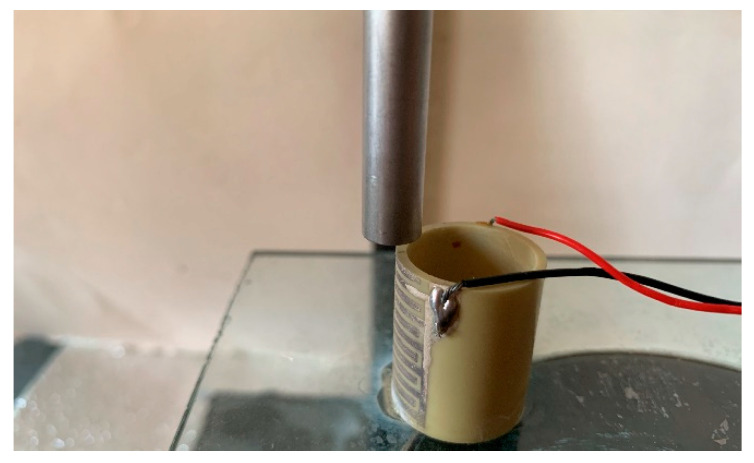
The laser head and sample.

**Figure 12 micromachines-13-01760-f012:**
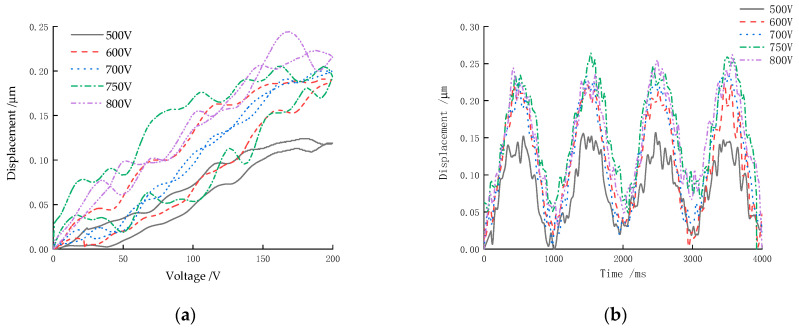
Displacement versus the polarization voltage: (**a**) response curves; (**b**) time-domain curves (four cycles).

**Figure 13 micromachines-13-01760-f013:**
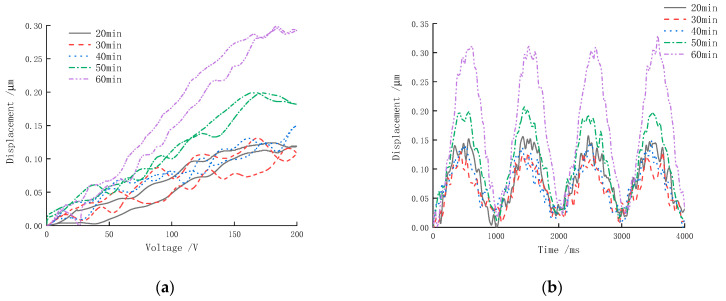
Displacement versus the polarization time: (**a**) response curves; (**b**) time-domain curves (four cycles).

**Figure 14 micromachines-13-01760-f014:**
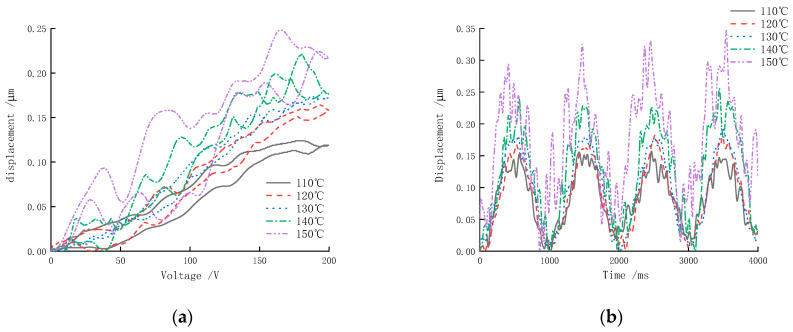
Displacement versus the polarization voltage: (**a**) response curves; (**b**) time-domain curves (four cycles).

**Figure 15 micromachines-13-01760-f015:**
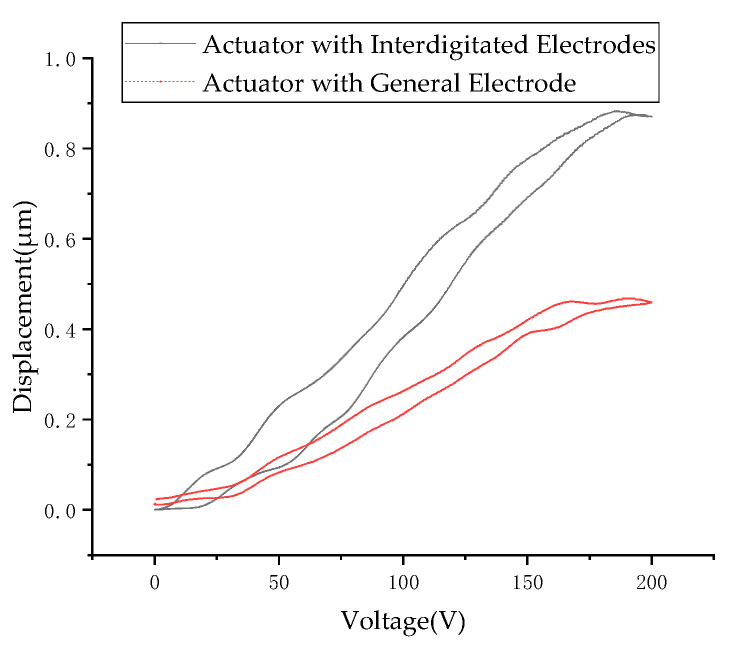
A plot of the output displacement versus the voltage.

**Table 1 micromachines-13-01760-t001:** Polarization process conditions.

Voltage/V	Temperature/°C	Time/min
500	110	20
600	120	30
700	130	40
750	140	50
800	150	60

## Data Availability

Data are available upon request from the corresponding author.

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
