# Peer review of "An Experimental Investigation on Polarization Process of a PZT-52 Tube Actuator with Interdigitated Electrodes"

_micromachines, 2022, doi:10.3390/mi13101760_

Round 1

Reviewer 1 Report

Dear Authors,

My comments are there in the attached file.

Major concerns:

1. FEM model

2. Novelty and recent models, any comparison or validation?

3. Research methodology need to be presented clearly.

4. Methods particularly the numerical side must be enhanced.

5. Abstract and conclusion needs to be double checked as per guidelines.

Regards,

Author Response

Response to Reviewer 1 Comments

Point 1: line 14 and line 21

Response: The abstract has been rewrited.

Point 2: line 77

Response: It has been revised.

Point 3: line 95-100

Response: They has been rewrited.

Lead Zirconate Titanate (PZT-52) is commonly used as an actuator material due to its high dielectric constant and piezoelectric constant. The raw materials were Pb(H2COO)2-3H2O (the purity is 99.5%), Ti((CH3)2CHO)4(98ï¼…), ZrO2 and C6H8O7H2O (99.5%). Pb(Zro.52Tio.48)O3.was used as the stoichiometric ratio and dopants such as La3+, Sb5+, Bi were added as well. It is relatively mature technology to prepare P-52 tubes by solid phase method, and the preparation process generally involves: ingredients – mixed grinding – pre-firing – second grinding – granulation– molding – plastic– sintering into porcelain – shape processing. Figure 2 shows the samples prepared in YU Hai Electronic Ceramics Co., Ltd.

Point 4: line 149: Why Solid 98? Did you check mesh independence?

Response: The electric field of the model is preliminarily analyzed and the region of maximum electric field is obtained. Then, 6 nodes SOLID 5,10 nodes SOLID 98 and 20 nodes SOLID 122 were used for electric field analysis. The local grid where the largest electric field is located is encrypted. For the same loading voltage, the difference of the maximum electric field strength is 5%.

There is no special purpose in using SOLID 98 for electric field analysis. Electromechanical coupling analysis of the model was performed using SOLID 98, which is an inertial operation.

Point 5: Figure 8: No explanation to FEM model and resuluts are no in accordance? Where is the mathematical model?

Response: The partial models were built using finite element analyze software ANSYS (ANSYS, Inc). SOLID98 ELEMENT was adopted for electric field analysis with 84103 nodes and 62345 units. The "0V" electrodes area is set to zero potential, and the "+" electrode area to high volts. A vector diagram of the axial electric field was shown in Figure 8.

There is non-uniformity and non-linearity in the electric field in the electrode attachment area. It is common for the electric fields to be uniform between branch electrodes. They are sparse in the area beneath the electrode layer. There is no unified mathematical model for electric fields. The mathematical model of electric field has no value for the overall polarization of the sample.

Point 6: Figure9: What does the strengh variance depict? Please put some explanation to the results.

Response: Calculations of electromagnetic fields are based on Maxwell's governing equations. An electrostatic field analysis by finite element method can be simplified as the Poisson equation of electric potential:

dφ=- d                                (1)

Where: d  is the distance between two points, dφ is the potential difference between two points, and  is the electric field vector. So, the maximum electric field strength of the sample is linearly related to the loading voltage. A piezoelectric tube actuator with general electrode is covered with electrodes on both inside and outside. It acts like a capacitor along the wall thickness and has uniform electric field strength, which is the voltage to wall thickness ratio. The electric field strength also increases linearly with increase of voltage.

Point 7: line 175

Response: It has been revised.

Reviewer 2 Report

1. the ceramic density can significantly influence the  piezoelectric performance could you give the density of the sintered ceramic samples?

2. please give more information and compare the effects of different polarization condition on piezoelectric performance of ceramic actuators

Author Response

Response to Reviewer 2 Comments

1.the ceramic density can significantly influence the  piezoelectric performance could you give the density of the sintered ceramic samples?

Response: Volume density is 7.6103 kg m-3 .

  1. please give more information and compare the effects of different polarization condition on piezoelectric performance of ceramic actuators

Response: There was a presentation of polarization and displacement data for PZT-52 tube actuators with general electrode in line 531-538 and Figure 15.

“In order to polarize the tube actuator with general electrode, it was heated to 150 °C for 40 min under 4kV voltage. In Figure 15, the axial displacement experiment is performed as the actuator with interdigitated electrodes. So, obtaining good polarization at relatively low voltages is of utmost importance from the functionality point of view. As can be seen from the figure, with the increase of driving voltage, the axial displacement gradually increases. Actuators with interdigitated electrodes can reach 0.88 μm, while actuators with general electrode can reach 0.47 μm.”

Reviewer 3 Report

Title- Experimental research on polarization process of piezoceramics tube actuator with interdigitated electrode

Authors: Y. Liu, A. Zeng, S. Zhang, R. Ma, Z. Du

Manuscript Id: 1941877

In this manuscript, the authors have fabricated the piezo actuator with interdigitated electrodes. The fabricated piezo actuator is tested for the actuation displacement in various environments including polarization voltage, time and temperature. After a careful review with interest, this reviewer feels that the manuscript is not suitable for publication in its current form. This manuscript also falls short of scientific writing and the English language to meet the standards of Micromachines. Nevertheless, authors are advised to revise the manuscript extensively, polish it for English/scientific language improvement, and hence recommend a major revision.

Below are some comments/suggestions for the authors;

[1] The authors should reconsider the title. It would be nice if it should be “An experimental investigation on the polarization process of piezoceramics tube actuator with interdigitated electrode”. This looks more scientific. Furthermore, authors should also consider the materials name/specifications (For ex. PZT in this case) in the title.

[2] Authors should also reconsider writing the abstract part. Lacks a scientific depth. Written in a more generic way. Again, the material specification/name is missing in the abstract. Which piezoceramics do the authors consider for this study? This makes the entire paragraph pointless. There are also many grammatical mistakes in the abstract.

[3] Page 1; abstract; line 18-20; Needs reconsideration. Rephrase the sentence. What are general piezoceramics? Please specify properly.

[4] All the abbreviations and acronyms should be properly defined when they are used for the first time in the manuscript. This will help in better readability. For ex., on page 1; introduction; line 30; what does MEMS stands for?

[5] Page 1; introduction; line 36-41 and Page 2; line 56; Repetitions. Does not make any sense. Please reconsider this.

[6] There is some confusion between the technical terms. Please verify. What is nanoscale micro-positioning? If it is nano or micro, then why authors are showing deformation in mm (i.e., 0.88 mm). It needs more clarification.

[7] The authors should reconsider the introduction section. What is the novelty of the current work? They should clearly mention. Again merely discussing literature (i.e., this author has done this and that) is not sufficient. What is the author's take on the previous literature? How do they connect with the current work? What are the gaps in previous work? They should clearly highlight these points in the introduction. Please refer to some important articles listed in Annexure 1 (a separate document) on piezoceramics. Try to look into it and refer.

[8] Which ceramics has been chosen for this study? What is the electrode material? Why PZT-52? What is its stoichiometric composition? How is PZT-52 advantageous over other ceramics (including Pb-based and Pb-free)? Is it synthesized by authors or directly procured from the supplier? How about the purity of oxide powders? How much is its Curie temperature? What is “a certain amount of binder” (page 3; line 98)? How much milling time? What are the parameters in the sintering process? Which hot-press machine and what about hot-pressing parameters? The experimental section should be rewritten. Needs more clarification. Currently, it seems superficial. Please refer to the articles in Annexure 1 for more clarity.

[9] The reviewer is curious about why upon applying the 800 V voltage, the breakdown field reaches to its maximum. Any comment? [10] It seems that the authors didn’t do justice to the results. Discussions are not adequate. For ex. Why the deformation is maximum for 800 V samples? They need to explain the physics/mechanisms. In the current form of the manuscript, it is lagging. Please consider this positively.

[11] A quick question; while the Curie temp. for PZT is around 200 °C, which means all the ferroelectric domains will start randomizing. In the present work, authors have argued that PZT samples exhibited maximum displacement at 150 °C, which seems contradictory. Please verify the tests. What about the repeatability of all the results? The thumb rule is that when the orientation of ferroelectric domains is randomized, they should exhibit less deformation, as the piezoelectric effect disappears. Any comments?

In my opinion;

[1] The work is a nice initiative for designing piezo actuators. However, experimental strategies are not well adopted in the work. Furthermore, the manuscript also falls short on the grounds of scientific discussions.

[2] Many unclear facts are presented in the manuscript, hence not acceptable in its present form. However, I advise authors to revise the manuscript extensively for possible consideration for publication.

Author Response

Response to Reviewer 3 Comments

[1] The authors should reconsider the title. It would be nice if it should be “An experimental investigation on the polarization process of piezoceramics tube actuator with interdigitated electrode”. This looks more scientific. Furthermore, authors should also consider the materials name/specifications (For ex. PZT in this case) in the title.

Response: The title has been changed to “An experimental Investigation on Polarization Process of a PZT-52 Tube Actuator with Interdigitated Electrodes”.

[2] Authors should also reconsider writing the abstract part. Lacks a scientific depth. Written in a more generic way. Again, the material specification/name is missing in the abstract. Which piezoceramics do the authors consider for this study? This makes the entire paragraph pointless. There are also many grammatical mistakes in the abstract.

[3] Page 1; abstract; line 18-20; Needs reconsideration. Rephrase the sentence. What are general piezoceramics? Please specify properly.

Response: The abstract has been rewrited.

“The manipulator is the key component of the micromanipulator. Using the axial expansion and contraction properties, the piezoelectric tube can drive the manipulator to achieve micro-motion positioning. It is widely used in scanning probe microscopy, fiber stretching and beam scanning. Piezoceramic tube actuator used to have continuous electrodes inside and outside. It is polarized along the radial direction. There are relatively high polarization voltages, but poor axial mechanical properties. A new tubular actuator is presented in this paper by combining interdigitated electrodes and piezoceramic tubes. The preparation, polarization and mesoscopic mechanical properties were investigated. Using Lead Zir-conate Titanate (PZT-52) as substrate, the preparation process of interdigitated electrodes by screen printing was studied. For initial polarization voltage determination, the local characteristic model of the actuator was extracted and the electric field was analyzed by a finite element method. By measuring the actuator's axial displacement, we measured the actuator's polarization effect. Various voltages, times, and temperatures were evaluated to determine how polarization affects the actuator's displacement. Optimal polarization conditions are 800 V, 60 min and 150 °C, with a maximum displacement of 0.88 μm generated by a PZT-52 tube actuator with interdigitated electrodes. PZT-52 tube actuators with continuous electrode cannot be polarized under these conditions. The maximum displacement is 0.47μm after polarization at 4kV. Based on the results, the new actuator has a more convenient polarization process and a greater axial displacement from an application standpoint. It provides technical guidance for the preparation and polarization of the piezoceramics tube actuator. There is potential for piezoelectric tubular actuators to be used in a broader range of applications.”

[4] All the abbreviations and acronyms should be properly defined when they are used for the first time in the manuscript. This will help in better readability. For ex., on page 1; introduction; line 30; what does MEMS stands for?

Response: All the abbreviations and acronyms have been defined.

[5] Page 1; introduction; line 36-41 and Page 2; line 56; Repetitions. Does not make any sense. Please reconsider this.

Response: The sentence in line36-41 has been modified as

“High-precision mechanical engineering, biomedical science, aerospace, and other micro-drive fields need high-quality actuators required both good electromechanical characteristics and low preparation cost[18-21].”

[6] There is some confusion between the technical terms. Please verify. What is nanoscale micro-positioning? If it is nano or micro, then why authors are showing deformation in mm (i.e., 0.88 mm). It needs more clarification.

Response: The 0.88mm is a mistake and has been changed to 0.88μm. The piezoceramic tube actuator is mesoscopic displacement in the paper.

[7] The authors should reconsider the introduction section. What is the novelty of the current work? They should clearly mention. Again, merely discussing literature (i.e., this author has done this and that) is not sufficient. What is the author's take on the previous literature? How do they connect with the current work? What are the gaps in previous work? They should clearly highlight these points in the introduction. Please refer to some important articles listed in Annexure 1 (a separate document) on piezoceramics. Try to look into it and refer.

Response: Thank you for providing 23 literatures to us. Some new literature reviews were added on line104-124 and line170-175.

[8] Which ceramics has been chosen for this study? What is the electrode material? Why PZT-52? What is its stoichiometric composition? How is PZT-52 advantageous over other ceramics (including Pb-based and Pb-free)? Is it synthesized by authors or directly procured from the supplier? How about the purity of oxide powders? How much is its Curie temperature? What is “a certain amount of binder” (page 3; line 98)? How much milling time? What are the parameters in the sintering process? Which hot-press machine and what about hot-pressing parameters? The experimental section should be rewritten. Needs more clarification. Currently, it seems superficial. Please refer to the articles in Annexure 1 for more clarity.

Response:

Lead Zirconate Titanate (PZT-52) is commonly used as an actuator material due to its high dielectric constant and piezoelectric constant. The raw materials were Pb(H2COO)2-3H2O (the purity is 99.5%), Ti((CH3)2CHO)4(98ï¼…), ZrO2 and C6H8O7H2O (99.5%). Pb(Zro.52Tio.48)O3.was used as the stoichiometric ratio and dopants such as La3+, Sb5+, Bi were added as well. It is relatively mature technology to prepare P-52 tubes by solid phase method, and the preparation process generally involves: ingredients – mixed grinding – pre-firing – second grinding – granulation– molding – plastic– sintering into porcelain – shape processing. Figure 2 shows the samples prepared in YU Hai Electronic Ceramics Co., Ltd.

Main material parameters of PZT-52: Dielectric constants,  , is 2400. Piezoelectric constant, d31 is 204x10-12C N-1 and d33 is 520 x10-12C N-1. Volume density is 7.6103 kg m-3. Curie point is 270 ºC. Young modulus is 59x109N m-2. Poisson ratio is 0.36.

The electrode material is conductive silver glue, which contains 60-70% silver (weight ratio) and a particle size of 3-5 μm, cured at 150°C.

[9] The reviewer is curious about why upon applying the 800 V voltage, the breakdown field reaches to its maximum. Any comment? [10] It seems that the authors didn’t do justice to the results. Discussions are not adequate. For ex. Why the deformation is maximum for 800 V samples? They need to explain the physics/mechanisms. In the current form of the manuscript, it is lagging. Please consider this positively.

Response: Through the electric field analysis in Section 2.3.1, the distribution cloud diagram of electric field is shown in Figure 8. There is non-uniformity and non-linearity in the electric field in the electrode attachment area. Near the electrodes, the electric field lines are relatively concentrated, and the electric field strength is relatively high. The maximum electric field strength appears at the marked MAX. However, in the middle area of the two branch electrodes, the electric field strength is relatively low. When 800V DC voltage is applied and the maximum electric field strength in the electrode area is close to 7kV/mm. The electric field in the middle area of the electrode is smaller.Therefore, taking the voltage of the maximum electric field strength as the reference basis to polarize the sample, the sample can be uniformly polarized. After all, the greater the applied DC voltage, the easier it is to make the domains turn to the direction of the electric field and fully polarize the sample.However, in the experiment, when 800V voltage polarization is loaded, the electrode area is easy to be broken down.

The black line is the breakdown field in the polarization experiment. This is the position near to the MAX electric field.

[11] A quick question; while the Curie temp. for PZT is around 200 °C, which means all the ferroelectric domains will start randomizing. In the present work, authors have argued that PZT samples exhibited maximum displacement at 150 °C, which seems contradictory. Please verify the tests. What about the repeatability of all the results? The thumb rule is that when the orientation of ferroelectric domains is randomized, they should exhibit less deformation, as the piezoelectric effect disappears. Any comments?

Response: The ratio of dopants and PZT-52 powder are controlled by YU Hai Electronic Ceramics Co., Ltd(http://www.zbyuhai.com). The Curie point of PZT-52 adopted in the paper is 270 ºC. Polarization temperature of the PZT-52 tubu actuator with continuous electrode is about 100~150°C in mass production. The sample polarization temperature was referred to this experience.

This paper focuses on the experimental study of the polarization process of the new actuator from macroscopic perspective. The effects of polarization voltages, temperatures and times on displacement are analyzed in mesoscale. A nanoscale analysis of the polarization process of actuators has not yet been performed because of the limited experimental conditions. The nature of actuator polarization in nanoscale has not been studied in domain engineering due to experimental limitations.

Round 2

Reviewer 1 Report

Recheck FEM for one more time and add better explanation. Rest looks good. The images needs to be checked.

Author Response

Response:

Figure 8 has been revised.

New Line:243-245

Additionally, the electrode edge area where the "MAX" is located is easily broken down in the polarization experiment.

Reviewer 3 Report

Title- An experimental Investigation on Polarization Process of a PZT-52 Tube Actuator with Interdigitated Electrodes

Authors: Y. Liu, A. Zeng, S. Zhang, R. Ma, Z. Du

Manuscript Id: 1941877R1

Manuscript has been revised substantially. Reviewer is almost satisfied with the comments addressed by the authors. However, still from the discussion point of view, manuscript falls short. Please shed light on the aspects related to ferroelectric domain activities and their role on the deformation behavior, as this is important in the deformation behavior of PZT. Following articles may be helpful. Authors are encouraged to go through them and enrich the discussion part. The final version of the manuscript needs to be amended mandatorily. There are some grammatical errors in the manuscript which can be easily fixed during proof-reading stage.  I recommend the acceptance of this manuscript to Micromachines and congratulate the authors for their nice contribution in the field.

https://doi.org/10.1016/j.actamat.2009.06.037

https://doi.org/10.1016/j.mtla.2021.101191

https://doi.org/10.1111/j.1551-2916.2010.04240.x

https://doi.org/10.1166/jcsmd.2018.1175

https://doi.org/10.1016/j.mtcomm.2021.102495

Author Response

Response:

New Line281-282:

For samples to exhibit piezoelectric behavior, all the domains need to be aligned in one direction by polarization [36].

New Line335-340:

There are some nonlinear fluctuations of curves primarily due to domain reorientation. The PZT-52 sample has defects such as porosity and impurities due to sintering [36]. The nonlinear contribution below the Curie temperature is primarily due to the motion of domain walls and their interactions with defects [37]. The ferroelectric domain configurations and defect vacancies can be controlled by the addition of aliovalent dopants in PZT-52, which causes a respective increase in the polarization and d33 values [38].
